# Interaction between Radiation Therapy and Targeted Therapies in HER2-Positive Breast Cancer: Literature Review, Levels of Evidence for Safety and Recommendations for Optimal Treatment Sequence

**DOI:** 10.3390/cancers15082278

**Published:** 2023-04-13

**Authors:** Kamel Debbi, Noémie Grellier, Gokoulakrichenane Loganadane, Chahrazed Boukhobza, Mathilde Mahé, Mohamed Aziz Cherif, Hanan Rida, Joseph Gligorov, Yazid Belkacemi

**Affiliations:** 1APHP—Radiation Oncology Department and Henri Mondor Breast Center, Henri Mondor University Hospital, 51 Avenue du Maréchal de Lattre de Tassigny, 94010 Créteil, France; 2Institut Mondor de Recherche Biomédicale (IMRB), INSERM U955, i-Biot, UPEC, 94000 Créteil, France; 3APHP—Medical Oncology Department, Institut Universitaire de Cancérologie, Sorbonne Université, 75013 Paris, France

**Keywords:** HER2 breast cancer, anti-HER2 targeted therapies, radiotherapy, safety

## Abstract

**Simple Summary:**

Anti-HER2 targeted therapies administered alone or in combination with chemotherapy have been extensively studied. However, only limited data are available regarding the safety of anti-HER2 therapies in combination with radiation. We propose a literature review of the risks and safety of combining radiotherapy with anti-HER2 therapies. We will focus on the benefit/risk ratio and try to understand the risk of toxicity in early-stage and advanced breast cancer. HER2-targeting monoclonal antibodies and checkpoint inhibitors can be combined with radiation, apparently with no excess toxicities. Considering the limited evidence, caution is required for combined radiation with TKI and antibody drugs.

**Abstract:**

**Purpose**: Over the past twenty years, anti-HER2 targeted therapies have proven to be a revolution in the management of human epidermal growth receptor 2 (HER2)-positive breast cancers. Anti-HER2 therapies administered alone or in combination with chemotherapy have been specifically studied. Unfortunately, the safety of anti-HER2 therapies in combination with radiation remains largely unknown. Thus, we propose a literature review of the risks and safety of combining radiotherapy with anti-HER2 therapies. We will focus on the benefit/risk rationale and try to understand the risk of toxicity in early-stage and advanced breast cancer. **Methods**: Research was carried out on the following databases: PubMed, EMBASE, ClinicalTrial.gov, Medline, and Web of Science for the terms “radiotherapy”, “radiation therapy”, “radiosurgery”, “local ablative therapy”, and “stereotactic”, combined with “trastuzumab”, “pertuzumab”, “trastuzumab emtansine”, “TDM-1”, “T-Dxd”, “trastuzumab deruxtecan”, “tucatinib”, “lapatinib”, “immune checkpoint inhibitors”, “atezolizumab”, “pembrolizumab”, “nivolumab”, “E75 vaccine”, “interferon”, “anti-IL-2”, “anti-IL 12”, and “ADC”. **Results**: Association of radiation and monoclonal antibodies such as trastuzumab and pertuzumab (with limited data) seems to be safe, with no excess risk of toxicity. Preliminary data with radiation and of antibody–drug conjugate of trastuzumab combined cytotoxic (trastuzumab emtansine, trastuzumab deruxtecan), given the underlying mechanism of action, suggest that one must be particularly cautious with the association. The safety of the combination of a tyrosine kinase inhibitor (lapatinib, tucatinib) and radiation remains under-studied. The available evidence suggests that checkpoint inhibitors can be safely administrated with radiation. **Conclusions**: HER2-targeting monoclonal antibodies and checkpoint inhibitors can be combined with radiation, apparently with no excess toxicities. Caution is required when associating radiation with TKI and antibody drugs, considering the limited evidence.

## 1. Background

The gene encoding human epidermal growth factor receptor 2 (HER2) is amplified and overexpressed in approximately 15–20% of patients with breast cancer. HER2 overexpression was associated with poor outcomes in the pre-trastuzumab era [1]. HER2-positive breast cancers are biologically more aggressive, with high cellular proliferation and metastatic rates in the absence of HER2-targeted therapies and more frequently associated with a worse prognosis compared to HER2-negative breast cancers. Moreover, HER2 overexpression predicts the response of HER2-targeting therapies, such as trastuzumab, pertuzumab, lapatinib, neratinib, tucatnib, trastuzumab emtansine (T-DM1), and trastuzumab deruxtecan (T-Dxd). The efficacy of these drugs have radically changed the natural history of HER2-positive breast cancer in adjuvant and metastatic settings.

In the early breast cancer setting, trastuzumab, pertuzumab, and trastuzumab emtasine are complicated by potential concurrent administration with radiotherapy. In view of the benefit obtained with these drugs and their half-lives, the strategy is based on concomitant administration, despite the absence of robust data in the literature. However, the majority of authors recommend compliance with drastic constraints on the heart at the time of irradiation.

In the metastatic breast cancer (MBC) setting, the longer survival of patients under anti-HER2 treatment raises the question of combination with radiotherapy for curative, analgesic, or palliative purposes. In fact, the question of sequence is still pending. The issue is that most HER2+ MBC patients who will go through disease progression while on treatment with approved HER2-targeted treatments, such as the antibody drug conjugate ado-trastuzumab (T-DM1), will need palliative radiation. Thus, a clear definition of the risk in combining radiation with these drugs is needed.

In addition, there are currently many options for new drugs, including immunotherapies such as checkpoint inhibitors (PD-1 and PD-L1 inhibitors), which have largely demonstrated their clinical benefit for the treatment of aggressive solid tumors [2,3]. In HER2+ breast cancer, the recent results of the KEYNOTE-014/PANACEA trial showed that 15% of the patients who had PD-L1+ tumors had partial responses to combined anti-PD-1 mAb (pembrolizumab) with trastuzumab without radiation [4]. Ongoing studies are currently testing pembrolizumab and high-dose fraction-directed tumors for the triple-negative subtype. High dose per fraction stereotactic radiotherapy is known to involve complex interactions between several components of the immune system to ensure cytotoxicity. In HER2+ breast cancer, some other treatments leading to immune-mediated cytotoxicity via passive (adoptive T-cell transfer and cytokine) or active (HER2-directed vaccine) mechanisms are also under development [5].

This review aims to summarize the reported data on combined radiotherapy delivered, either concomitantly or sequentially, to both passive and active immune therapies for the treatment of HER2+ breast cancer, either in adjuvant or metastatic settings. We will summarize practical recommendations and consensus opinions published in the last 20 years in adjuvant and metastatic settings and highlight perspectives for future developments.

In addition, in order to better stratify the testimony of this review and to define the side effects of anti-HER2 drugs associated with radiotherapy, we have summarized in Table 1 the potential organ-by-organ effects. The rest of the specific data will be detailed in each of the drug-specific sections.

## 2. Materials and Methods

Thanks to PubMed and Medline, several useful data published through October 14, 2022 were identified. We decided to focus on English and French articles for our research, using the following search terms: “radiotherapy”, “radiation therapy”, “radiosurgery”, “local ablative therapy”, and “stereotactic”, combined with “trastuzumab”, “pertuzumab”, “trastuzumab emtansine”, “TDM-1”, “T-Dxd”, “trastuzumab deruxtecan”, “tucatinib”, “lapatinib”, “Immune checkpoint inhibitors”, “atezolizumab”, “pembrolizumab”, “nivolumab”, “E75 vaccine”, “interferon”, “anti-IL-2”, “anti-IL 12”, and “ADC”. Titles and abstracts presented up to 14 October 2022 at major international meetings were screened to determine eligibility for the review. Case reports were also included. A CONSORT diagram has been designed to explain the review publication process and structure (Figure 1).

## 3. Results

HER2 directed MABS: Trastuzumab and Pertuzumab (Table 2).

### 3.1. Trastuzumab

Trastuzumab is the first humanized monoclonal antibody targeting the HER2 receptor, directed against the extracellular domain of the receptor and leading to blockage of the MAP kinase signaling pathways and the PI3K-Akt pathway, which diminishes the cell cycle, decreases tumor cell proliferation, and stimulates innate cellular immunity via ADCC.

**In vitro**, HER2+ cells showed greater radioresistance, HER2 activation being responsible for cell proliferation, less apoptosis, and greater ability to repair radiation-induced damage. If radiotherapy is used in conjunction with anti-HER2 therapies, the latter may contribute to increasing or restoring tumor radiosensitivity [7,8].

**In vivo,** overexpression of HER2 receptors in cardiomyocytes has been demonstrated, resulting in a cardioprotective effect [9,10,11,12].

Trastuzumab was first introduced in the management of patients with metastatic breast cancer following the publication of the phase 3 trial by Slamon in 2001 [1] and in an adjuvant situation following publications from Piccart (HERA) and Romond (NSABP B31 and N9831) in 2005, Joensuu (FinHer trial) in 2009, and Slamon (BCIRG 006) in 2011. Concomitant trastuzumab and radiotherapy are theoretically a risk, either in metastatic or adjuvant settings [12].

**In the metastatic setting**, the first study by Slamon in 2001 showed high rates of heart failure in cases of concomitant trastuzumab and chemotherapy, in particular with anthracyclines + cyclophosphamide 27% (including 16% of grade III and IV), compared to 13% in cases of trastuzumab + paclitaxel (2% severe) and 5% (2% severe) in the case of chemotherapy alone (anthracyclines (A) + cyclophosphamide (C) or paclitaxel) [9]. For trastuzumab and concomitant radiotherapy, the available data are of a low level of evidence. Two retrospective studies have shown efficacy of brain irradiation and concomitant trastuzumab. However, a case of possible radiation myelitis has been reported in the literature [13]. Therefore, it is necessary to remain cautious.

**In the adjuvant setting**, trastuzumab was started after the end of radiotherapy in the HERA and FinHer trials, whereas trastuzumab and radiotherapy were given concomitantly in BCIRG 006 and NSABP B31. There is no trial comparing trastuzumab and radiotherapy given concomitantly versus sequentially, but the data from BCIRG 006 and NSABP B31 provide information about the tolerance of concomitant trastuzumab and radiotherapy.

The phase III N9831 trial randomized 4 AC, then 12 paclitaxel versus 4 AC, then 12 paclitaxel, then weekly trastuzumab (52 cycles) versus 4 AC, then 12 paclitaxel + weekly trastuzumab (12 cycles), then weekly trastuzumab (40 cycles) [14]. Radiotherapy was delivered in 1503 of 2148 patients within 5 weeks after the last cycle of paclitaxel. In 74% of cases, the supraclavicular ± axillary lymph node areas were also irradiated. Internal mammary chain (IMC) irradiation, which was not authorized, was delivered to 44 patients (3%). There was no difference in the incidence of cardiac events, whether or not the patient was treated with radiotherapy (in the AC then paclitaxel + trastuzumab group: cumulative incidence of 1.7% in the RT group vs. 5.9% in the group without radiotherapy) and in cases of right versus left breast cancer. The 44 patients with IMC irradiation did not present an increased risk of a cardiac event or pneumonitis.

In the NSABP B31 trial, patients were randomized between AC then paclitaxel and AC then paclitaxel + trastuzumab (52 weeks). IMC irradiation was not allowed. There were more cardiac events in the trastuzumab group than in the control group (at 3 years: 4.1% vs. 0.8%, 95%CI 1.7–4.9%). However, patients treated with radiotherapy for left breast cancer did not have more congestive heart failure than those with right breast cancer (3.2% vs. 4%, *p* = 0.59, HR 0.8, 95%CI 0.38–1.7) [15].

Other prospective and retrospective studies have confirmed the safety of concomitant radiotherapy and trastuzumab [16,17]. However, given the potential cardiotoxicity of anthracyclines, trastuzumab, and radiotherapy, as well as the multiple other risk factors, authors often recommend limiting the mean heart dose as much as possible, either by IMC irradiation omission or by using more innovative radiotherapy techniques, such as arc therapy, helical tomotherapy, and/or deep inspiration breath hold for left-sided irradiation.

Aside from breast cancer, the safety of concurrent mediastinal irradiation, including the internal mammary area, and trastuzumab may be estimated from esophageal cancer irradiation data. The RTOG 1010 trial that investigated the impact of trastuzumab in -positive esophageal adenocarcinomas also provides some data on toxicity. This trial compared chemoradiotherapy (50.4 Gy + paclitaxel and carboplatin) with or without trastuzumab. The study showed that trastuzumab did not add overall grade ≥3 toxicity in the chemoradiotherapy vs. trastuzumab arm (69% vs. 79%). Importantly, there was no increased rate of grade 3 cardiac toxicity (4%), considering that radiation doses to the heart are much higher in esophageal cancer compared to breast cancer [18].

### 3.2. Pertuzumab

Pertuzumab is another humanized anti-HER2 monoclonal antibody. Its function is to bind to the extracellular dimerization domain of the HER2 protein; it then blocks ligand-dependent heterodimerization of HER2 and other HER family receptors, including EGFR (epidermal growth factor receptor), HER3, and HER4, thereby inhibiting ligand-dependent intracellular signaling pathways, in order to stop cell proliferation and apoptosis. Thus, the concomitant administration of these two antibodies strengthens the blocking of HER2 signaling pathways.

**In the metastatic setting**, the CLEOPATRA study demonstrated the benefit in terms of PFS and OS of pertuzumab associated with docetaxel + trastuzumab as first line therapy without additional cardiac toxicity [19,20]. The safety of concurrent radiation therapy was not investigated in this trial but was in a few retrospective studies, which reported good tolerance of these combined treatments [21,22].

A study from the Institut Curie assessed the combination of pertuzumab and trastuzumab with concomitant curative-dose locoregional breast radiotherapy in patients with metastatic breast cancer. Safety analysis did not reveal any significant adverse effects apart from 5.4% grade 3 radiodermatitis (5.4%), but no significant gastrointestinal or cardiac toxicity [23]. The follow-up in all these studies was short.

Three trials studied neoadjuvant pertuzumab NeoSphere (phase II) compared to docetaxel + trastuzumab vs. docetaxel + pertuzumab vs. docetaxel + trastuzumab + pertuzumab vs. trastuzumab + pertuzumab. Docetaxel + trastuzumab + pertuzumab showed improved complete pathological response (pCR) without increasing toxicity, particularly cardiac toxicity. However, pertuzumab was not continued after surgery and, therefore, not given concomitantly with radiotherapy [24]. In the TRYPHAENA (phase II) study, only trastuzumab was continued after surgery while pertuzumab was not [25]. GeparSepto (phase III) compared nab-paclitaxel versus paclitaxel in the neoadjuvant setting for early breast cancer. In HER2 patients, trastuzumab and pertuzumab were administered concomitantly with chemotherapy. Only trastuzumab was continued after surgery [26].

**In the adjuvant setting**, the phase III APHINITY trial studied the addition of pertuzumab to adjuvant chemotherapy and trastuzumab [27,28]. Pertuzumab showed an improvement in DFS without a change in OS, with a benefit in the subgroup of N positive patients. Radiotherapy was performed according to local recommendations, after chemotherapy but combined with anti-HER2 treatments. Primary cardiac events were rare in both groups (<1%).

**For concomitant radiotherapy** with trastuzumab and pertuzumab, several retrospective studies have shown the good tolerance of these combined treatments, either in loco-regional or distant metastatic radiotherapy [21,22,23]. However, given the short follow-up of most of these data, caution must be observed.

Regarding fractionation, a monocentric retrospective study by Sayan et al. [29] did not show any difference in cardiotoxicity in the 41 patients treated with trastuzumab and hypofractionated whole breast irradiation (WBRI) (42.56 Gy in 16 fractions or 36.63 Gy in 11 fractions + boost 13.32 Gy in 4 fractions) compared to 100 patients treated with conventional fractionated WBRI (50 Gy in 25 fractions). However, the lymph node irradiation (LNI) rate was higher in the conventional group compared to the hypofractionated group (75% vs. 11%). The heart mean dose was 1.63 Gy in the conventional group vs. 1.01 Gy in the hypofractionated group. Significant asymptomatic left ventricular ejection fraction (LVEF) decline was similar in the two groups (5% in the conventional group vs. 7% in the hypofractionated group, *p* = 0.203) [29].

In the FAST-Forward trial, trastuzumab was given concomitantly to WBRI in HER2-positive patients: 10% were treated with trastuzumab alone and 62–74% were treated with trastuzumab and chemotherapy [30]. However, there are no specific toxicity data in this subgroup. No patient had pertuzumab.

### 3.3. Trastuzumab Emtansine (T-DM1)

T-DM1, an HER2-targeted antibody and microtubule inhibitor conjugate, was validated in second line systemic therapy patients with HER2-positive MBC. It is also indicated in patients with residual invasive disease after neoadjuvant chemotherapy combined with trastuzumab. However, data on the safety of this anti-HER2 agent associated with radiotherapy are lacking.

The radiosensitization effect of combined T-DM1 and radiation has been reported in preclinical studies. In a xenograft gastric/esophagus HER2-positive cell model, Adams et al. reported that the concurrent administration of T-DM1 and irradiation on day 0 at a dose of 2.5 Gy over 3 consecutive days induced a higher response rate compared to radiation with trastuzumab [31]. In another study, Mignot et al. investigated the impact of different doses of irradiation in five human breast cancer cell lines presenting different levels of HER2 expression after prior exposure to T-DM1. Although a linear relationship between the level of HER2 expression and radioresistance was observed, T-DM1 was not a radiosensitizer under the experimental conditions of this study [32].

In clinical research, several retrospective and prospective trials have assessed the impact of the combination of trastuzumab and radiotherapy, but less is known about the association of TDM-1 with radiotherapy. T-DM1 is a well-known radiation sensitizer, along with other microtubule inhibitors such as taxanes and vinca alkaloids. Thus, a potential increase in the risk of radiation-induced adverse events (AEs), including dermatitis and pneumonitis, is possible.

Following the publication of the KATHERINE trial reporting a 50% relative reduction in the risk of recurrence or death, adjuvant T-DM1 has been become the standard of care for patients with residual disease after neoadjuvant chemotherapy with anti-HER2 therapy [33]. WBRI was offered to all patients undergoing breast-conserving surgery and those with locally advanced disease after mastectomy (clinical T3 or T4 disease and/or with clinical N2 or N3 disease).

Radiotherapy was initiated within 60 days after surgery. No subgroup analysis was made for patients undergoing radiation. However, some radiation-related AEs were reported. A moderate increase in the radiation-induced pneumonitis rate was observed in patients receiving T-DM1 (1.5%) compared to those receiving trastuzumab (0.7%). Similar rates of radiation-related cutaneous complication rates were noted in 25.4% and 27.6% of patients on the T-DM1 and the trastuzumab arms, respectively [27].

Given the potential cardiotoxicity associated with trastuzumab, a phase II trial investigated the cardiac safety and feasibility of WBRI with T-DM1 in a cohort of 116 patients (concurrent, *n* = 39; sequential, *n* = 77). No significant difference was observed between the two groups in terms of grade 3 toxicity. No grade 4 toxicity was reported from either treatment sequence [33]. No protocol-prespecified cardiac side effects or symptomatic heart failure events were reported after T-DM1 [34].

In a recent subgroup analysis of the KATHERINE trial, a rise of grade ≥3 toxicities was observed in the adjuvant irradiation group compared to the non-irradiated group (27.4% vs. 16.2%). The increase in grade ≥3 toxicities reported with T-DM1 included thrombocytopenia/anemia, skin radiation toxicity, hypokalemia, and diarrhea [33]. In a recent single-center experience, Zolcsak et al. reported the preliminary safety data of the concurrent administration of T-DM1 and radiotherapy in 14 patients with residual invasive HER2-positive breast cancer. Adjuvant breast or chest wall irradiation was delivered at a dose of 50 Gy in 25 fractions. While grade 1 radiodermatitis was the most common side effect, two patients developed a reversible grade 2 decrease in LVEF [35]. Considering the mechanism of action of T-DM1 (radiosensitization with microtubule inhibitors), and in the absence of strong safety data, concurrent radiation could be avoided, at least in cases where nodal irradiation is required.

In the MBC setting, the HER2-positive subtype is more likely than others to be associated with brain metastases, with an estimation of incidence of 50% in the course of the disease in metastatic HER2-positive breast cancer [36]. Regarding the tolerance of the concurrent administration of T-DM1 and brain irradiation, the little available evidence is from case reports or small series of patients. T-DM1 and concomitant whole-brain radiotherapy (WBRT) seems to be feasible, with no adverse reactions or any increase in clinically-significant toxicity. However, caution is needed with concurrent stereotactic radiosurgery (SRS), since several cases of complication have been reported in the literature. Carlson et al. reported one experience in 7 of 13 patients who received T-DM1 and SRS. They concluded that the high rate (57%) of clinical radiation necrosis (CSRN) was clearly unacceptable [37]. In another study on 45 patients with CNS metastases of breast cancer, Stumpf et al. found a 13.5-fold increase in the risk of developing radionecrosis when T-DM1 was combined with SRS [38].

Little evidence exists regarding the combination of T-DM1 and palliative bone radiotherapy. In 2016, Géraud et al. reported the results of concurrent radiotherapy and T-DM1 in three heavily pretreated patients with symptomatic bone metastases of HER2-positive BC. Using a hypofractionated radiotherapy schedule (15 Gy/5 fractions in two cases and 8 Gy/1 fraction in one case), all these patients experienced good symptomatic relief without any increase in toxicity [39]. In addition, T-DM1 was considered the standard of care for patients with HER2-positive MBC whose disease progressed after treatment with a combination of anti-HER2 antibodies and a taxane, based on the EMILIA trial [40].

### 3.4. Trastuzumab Deruxtecan (T-Dxd)

The DESTINY-Breast phase III trial compared the efficacy and safety of T-Dxd, an antibody–drug conjugate consisting of a humanized anti-HER2 monoclonal antibody linked to a topoisomerase I inhibitor payload, to T-DM1. T-Dxd improved both progression-free and overall survival compared to T-DM1. Given the fact that T-Dxd was associated with increased drug-related adverse events of grade 3 or 4 compared to T-DM1 (45.1% and 39.8%, respectively), extra caution is required with radiotherapy. Adjudicated drug-related interstitial lung disease or pneumonitis occurred in 10.5% of the patients in the T-Dxd group and in 1.9% of those in the T-DM1 group. There are limited data regarding the combination T-Dxd and radiation. Data suggest that regional breast irradiation increases the risk of pulmonary fibrosis. For example, in the EORTC study additional regional nodal irradiation was associated with an increased risk of pneumonitis compared to breast irradiation alone (4.4% in the nodal irradiation group vs. 1.7% in the control group, *p* < 0.001) [41]. Similarly, T-Dxd was associated with a significantly higher risk of pneumonitis. Moreover, gastrointestinal toxicities were more common with T-Dxd [42]. Therefore, caution is required with the combination of T-Dxd and thoracic or abdominal radiation.

For risk of CNS toxicity, the DEBBRAH phase II study showed that intracranial activity of T-Dxd and radiation is feasible with manageable toxicity in patients with HER2-positive and HER2-low breast cancer who received mainly WBRT and/or stereotactic radiosurgery (SRS). However, the authors did not mention the timing between irradiation and sequential administration of T-Dxd [43].

### 3.5. Tyrosine Kinase Inhibitors (TKIs)

#### 3.5.1. Lapatinib

Lapatinib, a 4-anilino-quinazoline, inhibits the EGFR (ErbB1) and HER2 (ErbB2) receptors, especially their intracellular tyrosine kinase domains. Lapatinib can reduce the growth of ErbB-dependent tumor cells in vitro and in various animal models. The association of lapatinib and trastuzumab could be synergistically effective, possibly without cross-resistance mechanisms [44].

The combination of lapatinib and radiation has been investigated for different irradiated volumes. One phase II study evaluated the feasibility of the combination of lapatinib and chest wall (CW) irradiation after surgery of locally advanced breast cancer, as well as the dose of lapatinib. Among the secondary objectives, it was possible to evaluate the impact of the oral EGFR/HER2 inhibitor on the signaling pathways of the receptors and downstream in the tumor tissue, as well as the correlations between the response and the inhibition of downstream signaling. Radiation therapy was given in fractions of 1.8–2 Gy, 5 days per week, up to a total dose of 35 to 70 Gy. Lapatinib was administered concurrently with chemoradiotherapy started 1 week after lapatinib [44].

A phase I study assessed the toxicity and safety of combining daily lapatinib with locoregional radiation therapy in patients with unresectable and locally recurrent or chemotherapy-refractory and locally advanced breast cancer. The combination of lapatinib and radiation was well-tolerated in this patient cohort, even though 7/19 patients presented grade 3 radiation dermatitis. No patient developed symptomatic cardiac dysfunction [45]. This study showed the safety of lapatinib administered at a dose of 1500 mg/day in combination with breast or chest wall irradiation [45].

Brain radiation associated with lapatinib has been extensively studied. The systematic review by Ippolito et al. including 1081 patients with HER2+ BC with brain metastases from nine studies showed safety of lapatinib and concurrent radiotherapy [46]. In the studies that addressed the impact of concurrent SRS and lapatinib, local control and overall survival were significantly improved. Unexpectedly, a significantly lower risk of radiation necrosis was reported with the association of lapatinib and SRS compared to SRS alone [47]. In summary, the combination lapatinib and radiation seems to be feasible and safe.

#### 3.5.2. Tucatinib

Tucatinib is a reversible, potent, and selective HER2-targeting tyrosine kinase inhibitor. It is a molecule that is >1000 times more selective for HER2 compared to the epidermal growth factor receptor (EGFR). Tucatinib can inhibit the phosphorylation of HER2 and HER3, resulting in downstream inhibition of cell signaling and cell proliferation, and induces HER2-stimulated tumor cell death. Tucatinib induces growth blockage of HER2-stimulated tumors. Its combination with trastuzumab showed greater anti-tumor activity in vitro and in vivo, compared to each of the drugs used alone [48].

To date, there are no studies specifically addressing the question of the safety of the concurrent administration of tucatinib and radiation.

The efficacy of tucatinib plus trastuzumab and capecitabine was evaluated in an international randomized (HER2CLIMB) study that included locally advanced unresectable or metastatic HER2-positive breast cancer, with or without brain metastases [49]. The population was largely pre-treated with trastuzumab, pertuzumab, and T-DM1. In patients with brain metastases (BMs), radiation therapy or surgery was not immediately required for the disease. While the oncological results were all improved in the tucatinib arm compared to the placebo arm, there were no data on radiation impact on BM patients who received tucatinib [49]. To our knowledge, HER2CLIMB is the only trial that included patients with irradiated BMs. The majority (75%) of the subgroup of patients with brain metastases were previously treated with WBRT, SRS, or surgery. Although the rate of radiation necrosis was unknown, there was no excess of headache in the tucatinib arm (21.3%) compared to the placebo arm (20.5%) [49]. These limited data cannot permit any statement regarding the safety of the combination of tucatinib and brain metastasis radiotherapy. On the other hand, given the concerns based on the overlapping gastrointestinal toxicities of tucatinib (diarrhea: 80.6%; nausea: 58.4%; vomiting: 35.9%), caution is needed when radiation involves the spine or the abdomen [48].

### 3.6. Immune Checkpoint Inhibitors

Immune checkpoint inhibitors targeting programmed cell death 1 (PD-1) and its ligand 1 (PD-L1) have disrupted the treatment and prognosis of various cancer models. The success of immune checkpoint inhibitors has drastically changed the landscape of cancer treatment. Numerous preclinical and clinical studies have suggested that combining immune checkpoint inhibitors with radiation could be a promising strategy for the synergistic enhancement of treatment efficacy. For example, the safety and effectiveness of pembrolizumab with concurrent hypofractionated radiation therapy was reported for lung and liver lesions from metastatic non-small cell lung cancer [50]. Since 2019, the use of atezolizumab + paclitaxel in patients with PD-L1+ triple-negative breast cancer (TNBC) has been validated following the results of a phase III trial (IMpassion 130) [51]. Similarly, studies combining radiation and immunotherapy were proven to be safe in locally recurrent metastatic triple-negative breast cancer in prospective studies [52,53,54]. The PANACEA phase 2 trial assessed the safety and anti-tumor activity of pembrolizumab associated with trastuzumab in advanced trastuzumab-resistant HER2-positive breast cancer. This combination was shown to be safe and showed activity with durable clinical benefit in patients with advanced, PD-L1 positive, trastuzumab-resistant HER2-positive breast cancer. The study did not report the feasibility of concurrent radiation. However, since radiation can be combined with both trastuzumab and pembrolizumab, we can speculate that radiation can be combined with trastuzumab and pembrolizumab concurrently [4]. Moreover, given the long half-lives of checkpoint inhibitors, the suspension of these therapies is inconvenient in clinical practice. Overall, concurrent SBRT and checkpoint inhibitors is safe, but close monitoring should be considered for patients receiving lung or abdominal SBRT [55].

### 3.7. Cytokine-Activated Mediation

The use of both interferon-gamma (IFN-γ) and anti-HER2 antibodies synergistically produces a reduction of tumor growth in HER2-expressing breast cancers [56]. In HER2+ metastatic breast cancer, IFN-γ was tolerable for the treatment of patients when combined with paclitaxel and HER2-targeted agents (pertuzumab and trastuzumab) [57]. Furthermore, we know that IL-12 and IL-2 can increase IFN-γ levels in mice with HER2+ breast cancer [58,59].

The association of conventional fractionated radiotherapy and interferon has been evaluated in many situations, such as melanoma, hemangioma, and bladder cancer. This association is safe [60,61]. Radiotherapy can also be associated with interleukin IL-2 or IL-12 [62].

### 3.8. E75 Vaccine

The nature of the E75 vaccine corresponds to a 9-amino acid human leukocyte antigen (HLA)-restricted peptide located in the HER2 extracellular domain. The E75 vaccine activates CD8+ and CD4+ Th1 responses and boosts immune response [63]. E75 vaccines have been studied and developed, especially in the adjuvant setting. The E75 vaccine was administered after the initial full management of HER2-positive localized breast cancers to prevent disease recurrence [64,65,66]. A study reported a subset of patients treated with E75 and radiation therapy. Although the radiation technique and doses were not mentioned, there was no evident additional toxicity with radiation [66].

### 3.9. Adoptive T Cell Therapies

Adoptive T cell therapies can include the genetic modification of T cell receptors using gene transfer technology. It is possible to recognize MHC I antigens with high affinity and create chimeric antigen receptor T cells (CARs) by fusing a costimulatory specific antibody protein to the endogenous T cell receptor; to infuse ex vivo expanded tumor-infiltrating lymphocytes; and to infuse peripheral ex-vivo tumor antigen-primed and tumor antigen-expanded T cells. These therapeutics have been studied more in onco-hematology [67].

It is possible to expand ex vivo T cells against HER2 in mice models. These cells have antitumor activity [68]. In a pilot trial, autologous HER2 T cells co-cultured with HLA2-peptide-loaded dendritic cells were used to treat one patient with HER2+ metastatic breast cancer [69].

The high efficacy of CAR T cells in treating patients with hematologic malignancies is promising for the treatment of solid tumors even though experience is limited.

One approach to enhancing its efficacy in solid tumors is to combine CAR T cell therapy with other treatment modalities, such as radiation, known to be associated with local and systemic immunomodulatory effects. A recently published review suggested that optimal dose, fractionation, and timing of radiation may play a critical role in enhancing CAR T cell therapy [70].

No data exist on the combination of adoptive T cell therapies and concurrent breast radiation therapy.

## 4. Discussion

In this work, we performed a comprehensive synthesis of the data published on the combination of radiation with the main systemic anti-HER2 therapies that are currently indicated in localized and metastatic HER2-positive breast cancer. The development of HER2-targeted therapies has dramatically improved the prognosis of patients with HER-positive breast cancer, both in the adjuvant and advanced settings. Adjuvant radiotherapy remains the standard of care after breast-conserving surgery and in patients with a high risk of locoregional relapse who undergo mastectomy. Table 3 presents a summary of the main data on safety of combined radiotherapy and anti-HER2 targeted therapies including recommendations for clinical practice.

It is important to note that increasing numbers of prospective studies are ongoing to investigate the combination of anti-HER2 therapies with radiation. However, only limited clinical data and a low level of evidence are available for pertuzumab. Ideally, it would certainly be pertinent to propose a multicenter randomized trial for each molecule to answer the question of the toxicity of these combinations. This is particularly important because the comparison of trials is not always obvious in the context of the considerable heterogeneity regarding radiotherapy parameters, drugs, dosages, and therapy sequence. Additionally, the evaluation of combined radiotherapy with new anti-HER2 therapies is complex because of the numerous factors involved. Regarding the dose and its fractionation, volumes and sites of treatment may have an impact on the outcome of a combination.

Curative-intent adjuvant radiotherapy may be offered concurrently with trastuzumab and T-DM1, as per the NCCN (National Comprehensive Cancer Network) guidelines. However, caution is required when considering irradiation of regional nodes and especially with irradiation of internal mammary nodes in left-sided breast cancers. The risk of toxicity is increased with T-DM1 compared to trastuzumab due to its mechanism of action. Some data suggest an unacceptable rate of brain radiation necrosis (50%) with SRS and concurrent T-DM1, and it should be avoided in the absence of solid evidence demonstrating their safety.

Regarding pertuzumab (always combined with trastuzumab) indicated in advanced disease, concurrent irradiation seems to be feasible and well tolerated, including in cases of locoregional breast irradiation. Larger studies and longer follow-up of current cohorts are needed to confirm the safety of this molecule with radiation.

The few data available on trastuzumab-Dxd and radiotherapy suggests a higher risk of toxicities in combination compared to the first generation of anti-HER2 monoclonal antibodies and warrant particular attention.

Regarding TKIs, lapatinib and tucatinib are validated in a third line systemic treatment in HER2-positive advanced disease. Data suggest that tolerance of radiation with lapatinib seems to be good. However, it should be noted that both lapatinib and tucatinib are supposed to be combined with capecitabine, which is a well-known radiosensitizer. Therefore, it is probably safer to offer a therapeutic window with the systemic therapy if radiation is indicated.

Even with the indication of palliative irradiation, sophisticated planning approaches such as intensity-modulated radiation therapy (IMRT) may be offered for lowering doses to OAR.

New image-guided and adaptive radiation methods are also emerging as feasible approaches to make radiotherapy more precise and effective.

A recent phase 3 trial reported a significant reduction in the incidence of symptomatic esophagitis with IMRT compared to standard conformal treatment in the palliative setting in advanced central lung tumors (2% vs. 24%, *p* = 0.004) [71].

## 5. Conclusions

Therapeutic advances in the management of HER2-positive breast cancer have led to a strategic revolution. They are based on the integration of monoclonal antibodies targeting HER2 with or without chemotherapy and tyrosine kinase inhibitors. The initial fear that “inhibition” of HER2 activity, especially in cardiomyocytes, could lead to cardiac toxicity of radiotherapy in the breast area and lymph nodes has now largely been overcome. The question of possible increased complications of radiotherapy at metastatic sites during treatment has also been largely resolved. However, we must remain cautious as new treatment strategies evolve. On the one hand, there are the new ADCs, whose payload and bystander effect raise the question of the safety of combining certain chemotherapies (topoisomerase I inhibitors) with radiotherapy; on the other hand are the new targeted therapies useful in controlling HR signaling pathways, such as CDK4/6 inhibitors, which could very probably be used in combination with anti-HER2 treatments in the metastatic setting.

## Figures and Tables

**Figure 1 cancers-15-02278-f001:**
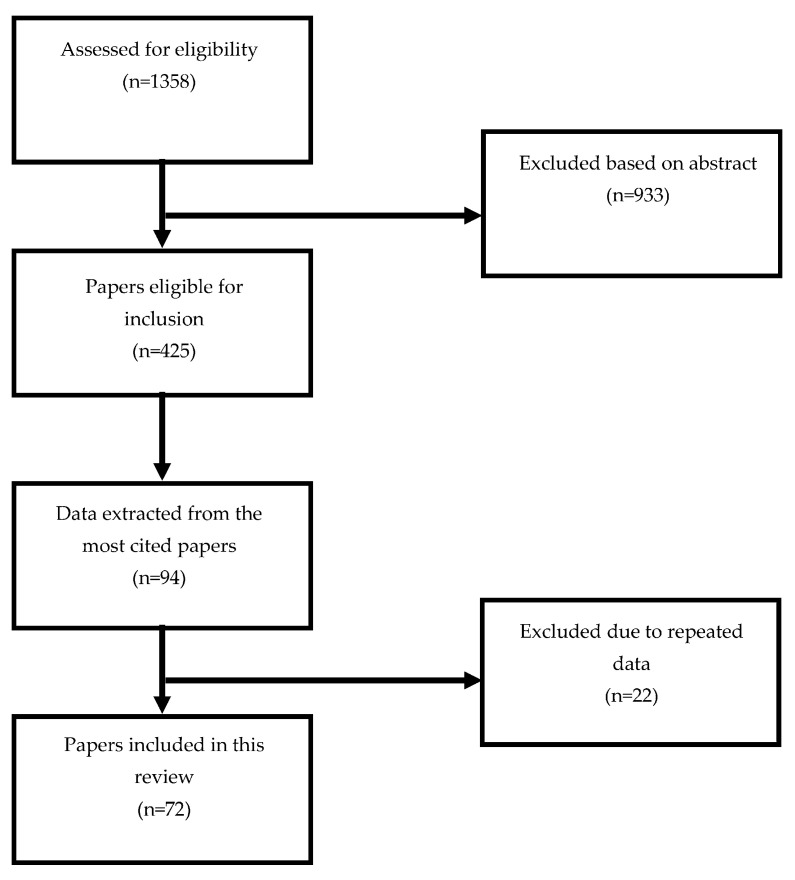
CONSORT diagram.

**Table 1 cancers-15-02278-t001:** Side effects incurred with the main targeted therapies in HER2-positive breast cancer.

	Cardiac	Lung	HM	Skin	GI	CNS
Trastuzumab	Yes	-	Yes	Yes	-	-
Pertuzumab	-	-	Yes	Yes	Yes	-
TDM-1	-	-	-	-	Yes	Yes
T-Dxd	-	-	Yes	-	Yes	Yes
Lapatinib	-	-	-	Yes	Yes	-
Tucatinib	-	-	-	Yes	Yes	-
Immune checkpoints inhibitors	Yes	Yes	Yes	Yes	Yes	Yes
Cytokine activated mediation	ND	ND	ND	ND	ND	ND
E75 vaccine	ND	ND	ND	ND	ND	ND

ND: No data; HM: hematological; GI: gastrointestinal; CNS: central nervous system; T-Dxd: trastuzumab deruxtecan.

**Table 2 cancers-15-02278-t002:** Guidelines, precautions, and advice for humanized monoclonal antibody (MAB) targeting HER2.

MAB	Half-Life (Days)	Concomitant Anti-Her2 and Radiotherapy	Guidelines	Precautions	Advice
Trastuzumab	28.5	Yes	NCCN: “Adjuvant HER2-targeted therapy ± endocrine therapy may be delivered concurrently with RT”. ESMO: “Trastuzumab may also be safely combined with RT and ET”.	Mean heart dose < 4 Gy (Quantec), 5 Gy As low as possible (DARBY NEJM 2013) Risk–benefit balance assessment of IMC irradiation	AVMI/tomotherapy DIBH [6]. Moderate hypofractionation possible for the whole breast. Conventional fractionation with lymph node irradiation for the moment.
Pertuzumab	18	Yes	ESMO: “It is recommended to decide on the administration of 1 year of trastuzumab/ Pertuzumab based on the risk assessment at diagnosis; the treatment may start before or after the surgery, in accordance with the approval wording by the regulators”.	Cautious in case of abdomen and pelvic irradiation that includes large GI volume	AVMI/tomotherapy DIBH [6]. No data on fractionation.

**Table 3 cancers-15-02278-t003:** Interaction between radiation therapy and targeted therapies in HER2-positive breast cancer: safety and recommendations for clinical practice.

	Breast	Breast + Nodal Irradiation	Palliative Radiation	WBRT	Brain Radiosurgery	Stereotactic Body Therapy (Excepted Brain)
Trastuzumab	Safe	Safe	Safe	Safe	Safe	Safe
Pertuzumab	Safe	Safe	Safe	Safe	Safe	Safe
TDM-1	Safe	Caution	Caution (in front of small bowel, discuss IMRT)	Safe	Caution (risk of radionecrosis)	Caution (with digestive organs)
Trastuzumab-Deruxtecan	Caution with hypofractionation	Caution with hypofractionation	Caution	Caution	Unknown	Unknown
Lapatinib	Safe	Caution	Safe	Safe	Safe	Safe
Tucatinib	Safe	Caution	Caution (in front of small bowel, discuss IMRT)	Safe	Unknown	Caution (with digestive organs)
Immune checkpoints inhibitors	Safe	Safe	Safe	Safe	Safe	Safe
Cytokine activated mediation	Safe	Safe	Safe	Safe	Safe	Safe
E75 vaccine	Caution	Caution	Caution	Caution	Unknown	Unknown

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
