# Peer review of "Interaction between Radiation Therapy and Targeted Therapies in HER2-Positive Breast Cancer: Literature Review, Levels of Evidence for Safety and Recommendations for Optimal Treatment Sequence"

_cancers, 2023, doi:10.3390/cancers15082278_

Round 1
Reviewer 1 Report
We thank the authors for investigating such an important topic.
They extensively reviewed targeted and immune treatments currently available or developed in patients with HER2-positive tumors.
However, as suggested by the paper’s title, we would expect more information on safety data relative to these treatments combined with radiation therapy. Information appears addressed for Trastuzumab, Pertuzumab, and T-DM1. But from T-DXd onwards, one remains hungry. For instance, in section 3.4, there is not at least a sentence informing that combination safety data is lacking. In the 3.5.1 section, the authors report the combination of lapatinib and chest wall irradiation but do not broach other irradiation sites described in the literature. I have the same comments for Tucatinib studies. The authors are laying around the beneficial effect of Tucatinib but do not provide safety data, if available, on the combination of this drug with radiation therapy.
On the other hand, the authors state that « in the context of breast cancer management, the combination of ICI and RT has been extensively studied in the literature » and only propose one reference that does not investigate this topic. In the same section, they give statements on the combination without any bibliography.
In the « Materials and Methods » section, the authors do not report the results of their literature search, which should appear at least as a CONSORT diagram.
Writing should be improved due to many mistakes and improper sentences.
Thus, this topic is interesting to investigate, but this article should be heavily revised before publication, focusing on the treatment combination.
Author Response
REVIEWER 1
Comment 1.
We thank the authors for investigating such an important topic. They extensively reviewed targeted and immune treatments currently available or developed in patients with HER2-positive tumors.
Response 1.
We thank the reviewer for this comment.
Comment 2.
As suggested by the paper’s title, we would expect more information on safety data relative to these treatments combined with radiation therapy. Information appears addressed for Trastuzumab, Pertuzumab, and T-DM1. But from T-Dxd onwards, one remains hungry. For instance, in section 3.4, there is not at least a sentence informing that combination safety data is lacking.
Response 2.
We thank the reviewer for this comment. There is limited data regarding the combination T-Dxd and radiation. Data suggests that regional breast irradiation increases the risk of pulmonary fibrosis. For example, in the EORTC study the additional regional nodal irradiation was associated with increased risk of pneumonitis compared to breast irradiation alone (4.4% in the nodal-irradiation group vs. 1.7% in the control group, P<0.001). Similarly, T-Dxd was associated with significantly higher risk of pneumonitis. Therefore, caution is required with the combination T-Dxd and radiation particularly when nodal RT is planned.
In addition to the details on lung potential toxicity, we added one paragraph for CNS risk of the combination between T-Dxd and brain irradiation. The paragraph is as follows:
“For the CNS risk of toxicity, the DEBBRAH phase II study showed that intracranial activity of T-Dxd and radiation is feasible with manageable toxicity in patients with Her2 positive and Her2 low breast cancer who received mainly whole brain radiation therapy (WBRT) and/or Stereotactic radiosurgery (SRS). However, the authors did not mention the timing between irradiation and sequential administration of T-Dxd [43]. »
All the changes are highlighted in green in section 3.4, page 9.
Comment 3.
In the 3.5.1 section, the authors report the combination of lapatinib and chest wall irradiation but do not broach other irradiation sites described in the literature.
Response 3.
Thank you for this pertinent comment. The combination of Lapatinib and radiation has been investigated in several situations and for different radiation volumes in the literature.
In the modified section 3.5.1, we pointed out data published in the chest wall irradiation, in the whole breast for unresectable and locally recurrent and locally advanced breast cancer settings. We also added one chapter on brain radiotherapy combined to Lapatinib. All the changes are highlighted in green in section 3.5.1, page 9-10.
Comment 4
I have the same comments for Tucatinib studies. The authors are laying around the beneficial effect of Tucatinib but do not provide safety data, if available, on the combination of this drug with radiation therapy.
Response 4
We thank the reviewer for this comment. We completely agree. The section 3.5.2 is changed including details on the HER2CLIMB trial. We also pointed out the details on brain metastases irradiation in terms of radionecrosis. For GI toxicity, while no report is published in patients receiving Tucatinib and abdomen or spine radiation we have based our recommendation for caution on the high rates of GI side effects of Tucatinib alone. The text is now adjusted accordingly and highlighted in green section 3.5.2, page 10, as follows:
“To our knowledge, HER2CLIMB in the only trial that included patients with irradiated BM. The majority (75%) of the subgroup of patients with brain metastases were previously treated with WBRT or SRS or surgery. Although the rate of radiation necrosis was unknown, there were no excess of headache in the Tucatinib arm (21.3%) compared to the placebo arm (20.5%) [45]. These limited data cannot allow any statement regarding the safety of the combination of Tucatinib and brain metastases radiotherapy. On the other hand, given the concerns based on overlapping gastrointestinal toxicities of Tucatinib (diarrhea : 80.6% ; nausea : 58.4% ; vomiting : 35.9%) caution is needed when radiation involves the spine or the abdomen[49].”
Comment 5.
On the other hand, the authors state that « in the context of breast cancer management, the combination of ICI and RT has been extensively studied in the literature » and only propose one reference that does not investigate this topic. In the same section, they give statements on the combination without any bibliography.
Response 5.
We thank the reviewer for this comment. We completely agree to adjust the references accordingly.
A large part of this chapter is dedicated to TNBC because ICI are indeed more developed in this subtype. Some ongoing trials are now testing Pembrolizumab + High dose RT-directed tumors in neo-adjuvant setting. In HER2 + patients’ data with concurrent RT are lacking. We found one reference from the PANACEA trial with however no details of radiation timing, parameters and toxicity. The text is added in section 3.6, page 10-11. The section 3.6 has been now completed.
Comment 6.
In the « Materials and Methods » section, the authors do not report the results of their literature search, which should appear at least as a CONSORT diagram.
Response 6.
Thank you for this important point. We have designed a CONSORT diagram (Figure 1). Il is now inserted in the “material and Methods section”, page 4.
Comment 7.
Writing should be improved due to many mistakes and improper sentences.
Response 7.
The English was corrected by an English native writer.
Comment 8.
Thus, this topic is interesting to investigate, but this article should be heavily revised before publication, focusing on the treatment combination.
Response 8.
Thank you. Hopefully, our responses and changes will be considered as satisfactory for the publication.
Reviewer 2 Report
In this manuscript, the authors are addressing a clinically relevant and indeed thoroughly discussed topic of breast cancer treatment. In the submitted review the authors show the potential risk of combined radiation therapy and targeted therapies in HER2-positive breast cancer.
The authors compile available literature about possible interactions and state potential risks at the end of their review.
Although the review is overall comprehensive some points need to be addressed before being considered for publication.
1. There are recent reviews around, tackling similar topics. The authors should state the uniqueness of their review e.g. compared to Kirova et al (10.3390/cancers13246358)
2. The introduction in general summarizes the problem of various targeted therapies and radiation. However, a pindown of the actual negative effect for patients would help a lot. The authors briefly state heart failure in the early setting and the interaction between several components of the immune system for metastatic breast cancer. A clear definition of the side effects of combined treatment would stratify the testimony of the review.
3. The structure of Trastuzumab and Pertuzumab is hard to follow at some points. A clear separation between neoadjuvant, adjuvant, and metastatic treatment would help a lot.
4. The chapter for T-Dxd is rather short and leaves out all the evidence and data from HER2 low patients.
5. The chapters about Tucatinib, E57 vaccination, and adoptive T-cell therapies are lacking a connection to radiotherapy.
6. The chapter about immune checkpoint inhibition states mainly TNBC and only briefly mentions HER2-positive breast cancer.
7. The chapter about Cytokine activation cites briefly studies in humans and mice. The impact of radiation and this therapy approach could be only shown in other tumor entities. At least an evaluation of the information is needed.
8. The authors mention a guideline and consensus for optimal treatment sequence in their title. To me, it was not clear where to find either of these in the manuscript. Clarification would be highly appreciated.
Author Response
REVIEWER 2
In this manuscript, the authors are addressing a clinically relevant and indeed thoroughly discussed topic of breast cancer treatment. In the submitted review the authors show the potential risk of combined radiation therapy and targeted therapies in HER2-positive breast cancer. The authors compile available literature about possible interactions and state potential risks at the end of their review. Although the review is overall comprehensive some points need to be addressed before being considered for publication.
Comment 1.
There are recent reviews around, tackling similar topics. The authors should state the uniqueness of their review e.g. compared to Kirova et al (10.3390/cancers13246358)
Response 1.
Compared to other review papers, the uniqueness of this extensive review is related to the HER2 subtype exclusively whereas the review by Kirova et al included all the BC subtypes. Moreover, we have reviewed here in addition to molecules targeting Her2 a larger number of other new molecules (such as immunotherapy) that were addressed specifically for HER2+ tumors.
Comment 2.
The introduction in general summarizes the problem of various targeted therapies and radiation. However, a pin down of the actual negative effect for patients would help a lot. The authors briefly state heart failure in the early setting and the interaction between several components of the immune system for metastatic breast cancer. A clear definition of the side effects of combined treatment would stratify the testimony of the review.
Response 2.
We thank the reviewer for this important comment. In the Background section we added one paragraph to point out the importance of this suggestion as follows:
“In addition, in order to better stratify the testimony of this review and to define the side effects of anti-HER2 drugs associated with radiotherapy, we have summarized in Table 1 the potential organ-by-organ effects. The rest of the specific data will be detailed in each of the drug specific sections”.
Also, we have constructed Table 1, to summarize the side effects that incurred with the main targeted therapies in HER2 positive breast cancer. Please see the changes that are highlighted in green.
Comment 3.
The structure of Trastuzumab and Pertuzumab is hard to follow at some points. A clear separation between neoadjuvant, adjuvant, and metastatic treatment would help a lot.
Response 3.
We thank the reviewer for this comment and understand very well the point with large chapter.
In this new version we made a clear distinction between metastatic and adjuvant/neoadjuvant settings in both chapters involving Trastuzumab and Pertuzumab. At the end of each chapter, we have focused the data of combined radiotherapy to both Trastuzumab and Pertuzumab. The changes are highlighted in green in this section page 6. We hope that this new structure of the chapters will meet your request.
Comment 4.
The chapter for T-Dxd is rather short and leaves out all the evidence and data from HER2 low patients.
Response 4.
We thank the reviewer for this comment. This point has been also proposed by Reviewer 1.
Indeed, there is limited data regarding the combination T-Dxd and radiation. Data suggests that regional breast irradiation increases the risk of pulmonary fibrosis. For example, in the EORTC study the additional regional nodal irradiation was associated with increased risk of pneumonitis compared to breast irradiation alone (4.4% in the nodal-irradiation group vs. 1.7% in the control group, P<0.001). Similarly, T-Dxd was associated with significantly higher risk of pneumonitis. Therefore, caution is required with the combination T-Dxd and radiation particularly when nodal RT is planned.
In addition to the details on lung potential toxicity, we added one paragraph for CNS risk of the combination between T-Dxd and brain irradiation. The paragraph is as follows:
“For the CNS risk of toxicity, the DEBBRAH phase II study showed that intracranial activity of T-Dxd and radiation is feasible with manageable toxicity in patients with Her2 positive and Her2 low breast cancer who received mainly whole brain radiation therapy (WBRT) and/or Stereotactic radiosurgery (SRS). However, the authors did not mention the timing between irradiation and sequential administration of T-Dxd [43]. »
All the changes are highlighted in green in section 3.4, page 9.
Comment 5.
The chapters about Tucatinib, E75 vaccination, and adoptive T-cell therapies are lacking a connection to radiotherapy.
Response 5.
We thank you for this comment.
Concerning Tucatinib, there are no studies addressing specifically the issue of concurrent radiation with Tucatinib. In this new version, we added all the data reported from the HER2CLIMB study which included patients with brain metastases who were previously treated with brain radiotherapy (WBRT or SRS or surgery). All changes are highlighted in green in section 3.5.2, page 10.
For E75 vaccination, one study reported a subset of patients treated with E75 and radiation therapy. Although the radiation technique and doses were not mentioned, there were no evident additional toxicity with radiation. These data now added in the section 3.8, page 11. Changes are highlighted in green.
For Adoptive T-cell therapies, there is no report on association with breast radiotherapy. We just found one reference that evokes a posible synergistic action of radiotherapy and car-T cell in the future. This is specified in section 3.9, page 11. Changes are highlighted in green.
Comment 6.
The chapter about immune checkpoint inhibition states mainly TNBC and only briefly mentions HER2-positive breast cancer.
Response 6.
We thank the reviewer for this comment.
We agree that a large part of this chapter is dedicated to TNBC. ICI are indeed more developed in this subtype. Some ongoing trials are now testing Pembrolizumab + High dose RT-directed tumors in neo-adjuvant setting. In HER2 + patients’ data with concurrent RT are lacking. We found one reference from the PANACEA trial with however no details of radiation timing, parameters and toxicity. The text is added in section 3.6, page 10-11.
Comment 7.
The chapter about Cytokine activation cites briefly studies in humans and mice. The impact of radiation and this therapy approach could be only shown in other tumor entities. At least an evaluation of the information is needed.
Response 7.
We thank the reviewer for this point.
In HER2+ breast cancer, no specific data exists for radiation and cytokine in humans but several studies were reported in other tumors sites such as melanoma, hemangioma, and bladder cancer involving radiation and cytokines. These data a re summarized in section 3.7, page 11.
Comment 8.
The authors mention a guideline and consensus for optimal treatment sequence in their title. To me, it was not clear where to find either of these in the manuscript. Clarification would be highly appreciated.
Response 8.
We thank the reviewer for this important remark.
It was not easy to give recommendations in the absence of high level of evidence in some cases. Thus, we decided to elaborate Table 2 which provides an overview of recommendations regarding the association of various molecules and radiation. Moreover, we changed the name of the tittle of the article “evidence for safety and Recommendations for optimal treatment sequence” to be less affirmative and omit confusion regarding the principles of Guidelines.
Reviewer 3 Report
Reviewer's report
Manuscript ID: Cancers 2238915
Title: Interaction between radiation therapy and targeted therapies in HER2 positive breast cancer: literature review, levels of evidence for risks, guidelines and consensus for optimal treatment sequence
Date: 2023/03/10
Reviewer's report:
This is an interesting manuscript as it’s a comprehensive review thru collected data from pubmed and medline aims to summarize the reorted data on combined radiotherapy delivered, either concomitantly or sequentially, to both passive and active immune therapies for the treatment of HER2+ breast cancer. The development of HER2-targeted therapies has dramatically improved the prognosis of patients with HER positive breast cancer, both in the adjuvant and advanced settings. Adjuvant radiotherapy remains the standard of care after breast-conserving surgery and in patients with high risk of locoregional relapse who undergo mastectomy. However, the late damage or potential toxicity was rarely discussed. This MS after thorough reviewed of various study and clinical trial show Her 2 targeting monoclonal antibodies and check-point inhibitors can be combined with radiation apparently with no excess toxicities I'm sure the result of this study could help in the decision-making process and guide towards an optimal management of HER-2+ breast cancer
The MS is well prepared and containing a large amount of data. Although, there remain some limitation. Nevertheless, it was still well written. Thus , it should be published.
Author Response
REVIEWER 3
Comment 1.
This is an interesting manuscript as it’s a comprehensive review thru collected data from pubmed and Medline aims to summarize the reported data on combined radiotherapy delivered, either concomitantly or sequentially, to both passive and active immune therapies for the treatment of HER2+ breast cancer. The development of HER2-targeted therapies has dramatically improved the prognosis of patients with HER positive breast cancer, both in the adjuvant and advanced settings. Adjuvant radiotherapy remains the standard of care after breast-conserving surgery and in patients with high risk of locoregional relapse who undergo mastectomy. However, the late damage or potential toxicity was rarely discussed. This MS after thorough reviewed of various study and clinical trial show Her 2 targeting monoclonal antibodies and check-point inhibitors can be combined with radiation apparently with no excess toxicities. I'm sure the result of this study could help in the decision-making process and guide towards an optimal management of HER-2+ breast cancer
The MS is well prepared and containing a large amount of data. Although, there remain some limitation. Nevertheless, it was still well written. Thus, it should be published.
Response 1.
We warmly thank the reviewer for this kind words and support for publication.
Round 2
Reviewer 1 Report
We thank the authors for their extensively revised version. Remarks and demands have been addressed. Still, the text writing requires a check since some sentences are unreadable.
As an example, in the discussion section:
"In this work, performed a comprehensive synthesis of knowledge on the combination of radiation and main HER2 + systemic therapies indicated in localized or metastatic Her2 513 + breast cancer, we."
Author Response
Comments and Suggestions for Authors from Reviewer 1
We thank the authors for their extensively revised version. Remarks and demands have been addressed. Still, the text writing requires a check since some sentences are unreadable.
As an example, in the discussion section:"In this work, performed a comprehensive synthesis of knowledge on the combination of radiation and main HER2 + systemic therapies indicated in localized or metastatic Her2 513 + breast cancer, we."
Response
We thank again reviewer 1 for the comment regarding the Language revision. English has been revised by Ms Myrna Perlmutter, a native English speaker and secretary of the Radiation Therapy department, Haifa University, Israel. Thus, we added her name in the Acknowledgments line 802:
“Acknowledgments: We thank Ms Myrna Perlmutter for her help to revise English of the manuscript”
All these language changes/corrections are left in a “track changes” form in the “non-cleaned” version
We also corrected the sentence line Number 513 as follows:
“In this work, we performed a comprehensive synthesis of the data published on the combination of radiation to the main anti-HER2 systemic therapies that are currently indicated in localized and metastatic Her2 + breast cancer.”
Reviewer 2 Report
The authors have thouroughly revised their manuscript and have addressed all the points I have raised before. Importent passages have been reworked and the inclusion of new comprehensive tables makes the review overall good to read and certainly interesting for the community.
Author Response
Comments and Suggestions for Authors from Reviewer 2
The authors have thoroughly revised their manuscript and have addressed all the points I have raised before. Important passages have been reworked and the inclusion of new comprehensive tables makes the review overall good to read and certainly interesting for the community.
Response
We thank again reviewer 1 for the comment